# Effects of Biochar on Container Substrate Properties and Growth of Plants—A Review

**Lan Huang [1] and Mengmeng Gu [2],*** 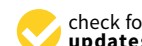

[1]    Department of Horticultural Sciences, Texas A&M University, College Station, TX 77843, USA;
       huanglan92@tamu.edu
[2]    Department of Horticultural Sciences, Texas A&M AgriLife Extension Service,
       College Station, TX 77843, USA
*     Correspondence: mgu@tamu.edu; Tel.: +1-979-845-8567

**Abstract:** Biochar refers to a processed, carbon-rich material made from biomass. This article provides a brief summary on the effects of biochar on container substrate properties and plant growth. Biochar could be produced through pyrolysis, gasification, and hydrothermal carbonization of various feedstocks. Biochar produced through different production conditions and feedstocks affect its properties and how it performs when incorporated in container substrates. Biochar incorporation affects the physical and chemical properties of container substrates, including bulk density, total porosity, container capacity, nutrient availability, pH, electrical conductivity and cation exchange capacity. Biochar could also affect microbial activities. The effects of biochar incorporation on plant growth in container substrates depend on biochar properties, plant type, percentage of biochar applied and other container substrates components mixed with biochar. A review of the literature on the impact of biochar on container-grown plants without other factors (such as irrigation or fertilization rates) indicated that 77.3% of the studies found that certain percentages of biochar addition in container substrates promoted plant growth, and 50% of the studies revealed that plant growth decreased due to certain percentages of biochar incorporation. Most of the plants tested in these studies were herbaceous plants. More plant species should be tested for a broader assessment of the use of biochar. Toxic substances (heavy metals, polycyclic aromatic hydrocarbons and dioxin) in biochars used in container substrates has rarely been studied. Caution is needed when selecting feedstocks and setting up biochar production conditions, which might cause toxic contaminants in the biochar products that could have negative effects on plant growth.

**Keywords:** container substrates; physical properties; chemical properties; biomass

## 1. Introduction

Biochar refers to processed, carbon-rich material derived from biomass [1–3]. Recent research has shown that biochar can be used as a replacement for commonly-used container substrates [4–8]. Container substrates are often soilless, making it easy to achieve consistency. Primary substrate components include peat moss, vermiculite, perlite, bark, and compost [9]. Peat moss is an excellent substrate component; it has essential characteristics such as low pH, high cation exchange capacity (CEC), and appropriate aeration and good container capacity [10–12], which are ideal for horticultural container application. However, intensive extraction of peat from peatlands can damage natural habitats and release $CO_2$ into the atmosphere if the disturbed peatland is left unrestored [13]. The United Kingdom government has thus proposed reducing the use of peat [14]. The cost of this commonly-used substrate is also high due to the extreme cost of transportation, fuel for extraction, and processing [9,15]. Therefore, it is beneficial and necessary to search for alternative



environmentally-friendly and local substrate components [9,16]. Research has shown that biochar could be a potential alternative to commonly-used substrates. Using biochars (a byproduct of bioenergy production) in agriculture adds value to bioenergy production [17]. Biochar could offer economic advantages over other commonly-used substrates, if produced on site. Extensive research has shown that replacing a certain percentage of commonly-used container substrates with biochar could increase plant growth in certain conditions [18–22].

However, biochars are variable, and their impact on container substrates could vary. It would be of interest to examine the characteristics of biochars, their incorporation in container substrates, and their effects on diverse types of container-grown plants. In this review, we provide a brief summary of the effects of biochar on container substrate properties and plant growth, and discuss the potential mechanism behind their effects. This review examines factors related to the impact of biochar, which include feedstock sources, production conditions, percentage of biochar applied, other substrate components mixed with biochar, and plant species. These factors can help address the general hypothesis that incorporation of biochar may not always have beneficial effects on container substrate properties or plant growth.

## 2. Biochar Production

There are many variables prior to, during, and after production of biochar. These factors will eventually affect biochar properties and its effect on plant growth and container properties when incorporated in container substrates.

### 2.1. Biochar Production Methods

There are three main processes to produce biochar: pyrolysis, gasification and hydrothermal carbonization. Pyrolysis is the thermal decomposition of biomass by heating (around 400 °C to 600 °C) without oxygen [23–25]. Compared to pyrolysis, gasification is conducted under small amounts of oxygen at relatively higher temperatures (around 700 °C to 1200 °C) [2]. Gasification produces smaller quantities of biochar with lower carbon (C) content than pyrolysis [2,25,26]. Hydrothermal carbonization uses water and catalysts at lower temperatures (180 to 300 °C) under high pressure to convert biomass to a different type of biochar product, hydrochar [27,28]. Hydrochars are acidic, and have low surface areas, less aromatic compounds, and higher CEC than those produced by pyrolysis and gasification [28,29]. Production temperature significantly influences the characteristics of biochars (Table 1). Biochar made from pruning waste at 500 °C had higher pH and different container capacity, total porosity, electrical conductivity (EC) and CEC, when compared to biochar produced at 300 °C [20]. Biochars made from different production processes can have different physical and chemical properties.

Utilizing biochar in agriculture adds values to biomass pyrolysis and gasification. The main purpose of fast pyrolysis is to produce syngas and bio-oil [17,23] and gasification syngas [30], with biochar being the byproduct. Syngas mainly includes carbon monoxide and hydrogen [31]. It could be used to provide energy for other pyrolysis processes. Bio-oil could be burned to produce heat or further processed to be used as fuel [32]. A specific process and its heating rate could be modified to produce desirable products. For example, gasification has higher yields of syngas and energy than pyrolysis [33]. Liquid bio-oil produced by pyrolysis has higher energy density and is cheaper and easier to transport; however, it is corrosive, which makes it difficult to store for a long time [31]. Slow pyrolysis produces more biochar and syngas, and fast pyrolysis more bio-oil [34]. The residence time (the amount of time taken in the pyrolysis procedure) of slow pyrolysis is from 5 min to 30 min, while that of fast pyrolysis is from seconds to less than a second, and the temperatures are higher [23,35]. Raising pyrolysis temperature can decrease biochar yield [36,37]. Production conditions can be adjusted on the basis of whether the desirable products are biochar or bioenergy products (bio-oil or syngas). Low temperatures and slow pyrolysis could be used to produce more biochar than the other products.

Pre-treatment of feedstocks has been reported to have a significant influence on biochar ash content, yield, and properties. Pre-treatment of biomass, such as washing with water or acid, could help remove

some ash culprits in feedstocks to reduce fouling, and improve the quality of biomass feedstock and final biochar products [37,38]. Rahman et al. [37] tested the effectiveness of different pre-treatments by comparing the EC of the initial washing medium and leachate collected after treatments. The result showed that the leachate EC of palm kernel shell increased, when pre-treated with dilute acid, dilute alkali, and distilled water. The highest increase in EC was found using dilute acid pre-treatment as a result of the removal of soil and alkaline metal by the acid solution and degradation of the biomass chemical composition. The ash content of palm kernel shell was reduced when pretreated with distilled water or diluted acid. The ash content increased with alkaline pre-treatment since abundant sodium ions in alkaline medium prevented ions from leaching into the medium and ions were bound and tied up by the biomass particles, which resulted in a high amount of ash content [37]. Torrefaction pre-treatment, which is a low temperature thermal conversion conducted without oxygen aiming to reduce moisture content of the biomass, could increase biochar yield during pyrolysis because the pretreatment predisposed carbon and oxygen content to remain as solids [39]. Another study showed that paper mill sludge as biochar feedstock, pre-treated with phosphoric acid and torrefaction, followed by pyrolysis, resulted in reduced volatile matter content, increased inorganic matter, and increased biochar yield [40]. It was shown that biochar made from feedstock with pretreatments such as light bio-oil or phosphoric acid may have larger surface areas and more porous structure [41,42], which could influence the effects of biochar on air space, nutrient and water-holding ability, and microbial activity. Biochar made from bark pre-treated with tannery slurry as an alkaline treatment could have a higher $NH4^+$ absorption capacity, as well as more surface functional groups (carboxyl and carbonyl groups) formed than untreated ones [43], causing increased CEC of biochar. Silica enrichment was also found in biochar made from rick husk pretreated with bio-oil or HCl [42].

In addition to pre-treatments of feedstocks, post-treatments could also change biochar properties. Some biochars could contain toxic compounds such as polycyclic aromatic hydrocarbons (PAHs) during production. Drying biochars at temperature of 100 °C, 200 °C, and 300 °C significantly decreased the amount of PAHs in biochars, which indicated that the release of PAHs from biochars was due to the increased opening of the pores and diffusion of PAHs from the pores after the thermal treatment [44]. Biochar could be treated and mixed with other substances. Dumroese et al. [7] dry-blended biochar with wood flour, polylactic acid, and starch to form pelleted biochar, which is preferred over the original fine-textured and dusty form for its handling convenience and even incorporation. McCabe et al. [45] evenly blended soybean-based bioplastics with biochar in a pelletized form as a source of nutrients in container substrates.

### 2.2. Biochar Feedstocks

In addition to production conditions, biochars could be made from varying feedstocks, which would contribute to differences in physical and chemical properties (Table 1). The feedstocks could be waste materials such as green waste [18], forest waste [46,47], wheat straw [5], sugarcane bagasse [48], rice hull [49], crab shell [50] and *Eucalyptus saligna* wood chips (byproduct of construction, fuel-wood and pulp wood) [51]. Biochars could also be made from non-waste materials such as holm oak [52], conifer wood [53], citrus wood [54] and pine wood [6,55–57]. The crab shell biochar and oak chip biochar have different pH ECs, and C, nitrogen (N), phosphorus (P) and potassium (K) content, although they were made by the same production method and temperature [50]. Straw biochar had a higher pH, exchangeable cations, and K content compared to a wood biochar [5]. The biochars made from sewage sludges of two different municipal plants also had slightly different pHs and N content [19]. Biochar could have high P and K content and could be used as P and K fertilizers, when made from rice hulls with high content of the minerals [49]. It was shown that biochar properties were related to the properties of the original feedstock [49]. The biochars made from different feedstocks could have different physical and chemical properties, which should be taken into account when they are incorporated in containers.

**Table 1.** Summary of the feedstock, production condition and properties of the biochars used in container substrates.

| Biochar Feedstock | Production Temp (°C) | CC (%) | AS (%) | TP (%) | BD (g cm$^{-3}$) | pH | EC (dS m$^{-1}$) | CEC (cmol kg$^{-1}$) | N (%) | C (%) | P (%) | K (%) | Na (%) | Ca (%) | Mg (%) | S (%) | Reference |
|---|---|---|---|---|---|---|---|---|---|---|---|---|---|---|---|---|---|
| Citrus wood [z] | n | n | n | n | n | 7.6 | 1.6 | n | 0.6 | 70.6 | 0.0008 | 0.37 | 0.32 | 0.02 | 0.01 | 0.07 | [54] |
| Coir (coconut husk fiber) | 450 | 64 | 33 | 97 | 0.14 | 8.2 | 1.0 | 153 | 1.3 | n | 0.17 | 1.89 | 4.83 | 0.33 | 0.73 | n | [21] |
| Conifer wood | 450 | n | n | 92 | 0.64 | 8.5 | 0.4 | n | n | n | n | n | n | n | n | n | [53] |
| Crab shell [z] | 200–250 | n | n | n | n | 8.8 | 0.005 | n | 3.6 | 28.7 | 0.03 | 0.61 | 0.04 | 0.18 | 0.08 | n | [50] |
| *Eucalyptus saligna* wood chip | 550 | n | n | n | n | 8.8 | 0.2 | n | 0.3 | 83.6 | 0.02 | 0.24 | n | 2.13 | 0.11 | 0.05 | [51] |
| Forest waste | n | n | n | n | n | 9.6 | 0.7 | n | 0.7 | 59.5 | 0.08 | 0.87 | 0.04 | 2.90 | 0.24 | 0.07 | [46,47] |
| Green waste | 550 | n | n | n | n | 7.7 | n | 250 | 0.3 | 77.5 | n | n | n | n | n | 0.00 | [58] |
| Green waste (willow, pagoda tree and poplar) | n | 27 | 22 | 49 | 0.44 | 8.0 | 0.9 | n | 1.2 | 50.4 | 0.01 | 0.47 | n | n | n | n | [18] |
| Green waste (tomato crop) | 550 | n | 28 | n | 0.13 | 10.4 | 3.3 | 524 | n | 55.0 | n | n | n | n | n | n | [59] |
| Hardwood pellets | n | n | n | n | 0.38 | 8.0 | 1.1 | n | n | n | 0.0005 | 0.04 | 0.001 | 0.02 | 0.002 | n | [5] |
| Holm oak | 650 | 51 | 29 | 80 | 0.32 | 9.3 | 0.5 | n | 0.9 | n | 0.18 | 0.77 | n | 3.76 | 0.40 | n | [52] |
| Mixed hardwood (oak, elm, and hickory) | 450 | n | n | n | 0.28 | n | n | n | n | n | 0.29 | 3.59 | 0.02 | 38.28 | 0.97 | n | [60] |
| Mixed hardwood | n | 60 | 24 | 85 | 0.15 | 11.2 | 2.0 | n | 0.2 | n | 0.05 | 0.64 | 0.01 | 2.75 | 0.13 | 0.02 | [61] |
| Mixed softwood [y] | 800 | n | n | n | n | 10.9 | 0.5 | 19 | n | n | 0.02 | n | n | n | n | n | [62] |
| Oak chip [z] | 200–250 | n | n | n | n | 5.1 | 0.3 | n | 0.1 | 52.2 | 0.09 | 0.10 | 0.06 | 1.03 | 0.08 | n | [50] |
| Olive mill waste | 500 | n | n | n | n | 9.7 | 9.2 | n | 0.6 | 59.5 | 0.90 | 6.42 | 0.05 | 3.40 | 0.61 | 0.17 | [47] |
| Pine chip [z] | 200–250 | n | n | n | n | 6.4 | 0.03 | n | 0.3 | 53.7 | 0.05 | 0.65 | 0.05 | 0.23 | 0.08 | n | [50] |

**Table 1.** *Cont.*

| Biochar Feedstock | Production Temp (°C) | CC (%) | AS (%) | TP (%) | BD (g cm$^{-3}$) | pH | EC (dS m$^{-1}$) | CEC (cmol kg$^{-1}$) | N (%) | C (%) | P (%) | K (%) | Na (%) | Ca (%) | Mg (%) | S (%) | Reference |
|---|---|---|---|---|---|---|---|---|---|---|---|---|---|---|---|---|---|
| Pine cone [z] | 200–250 | n | n | n | n | 5.1 | 1.2 | n | 0.6 | 53.2 | 0.01 | 0.16 | 0.04 | 0.36 | 0.05 | n | [50] |
| Pine wood | 450 | 49 | 34 | 83 | 0.17 | n | n | n | 0.4 | 48.1 | n | 0.10 | n | 0.50 | 0.30 | n | [6,57] |
| Pine wood | 450 | 47 | 36 | 83 | 0.18 | 5.4 | 0.2 | n | n | n | n | n | n | n | n | n | [55,56] |
| Poplar [y] | 1100–1200 | 57 | 34 | 91 | n | 9.7 | 0.2 | n | 0.7 | n | 0.51 | 0.98 | n | 4.31 | 7.64 | n | [63] |
| Pruning wastes | 300 | 17 | 4 | 21 | 0.18 | 7.5 | 0.3 | 26 | 1.2 | 66.2 | 0.004 | n | n | n | n | n | [20] |
| Pruning wastes | 500 | 35 | 4 | 39 | 0.18 | 10.3 | 1.0 | 16 | 1.2 | 77.7 | 0.01 | n | n | n | n | n | [20] |
| Rice husk | 500 | n | n | n | n | 10.2 | 0.8 | 50 | 0.3 | 20.5 | n | n | n | n | n | n | [64] |
| Rice husk [z] | n | n | n | n | 0.30 | 7.3 | n | n | 1.1 | n | 0.10 | 0.50 | n | n | n | n | [65] |
| Rice husk [z] | 200–250 | n | n | n | n | 6.3 | 0.4 | n | 0.6 | 45.4 | 1.21 | 0.27 | 0.73 | 15.80 | 1.04 | n | [50] |
| Rice hull [y] | 815–871 | n | n | n | 0.20 | 10.5 | n | n | 0.2 | 17.7 | 0.30 | 0.98 | n | 0.35 | 0.15 | 0.03 | [49,66,67] |
| Sewage sludge [x] | 450 | n | n | n | n | 7.9 | 1.1 | n | 1.1 | n | n | n | n | n | n | n | [19] |
| Sewage sludge [x] | 450 | n | n | n | n | 7.5 | 1.1 | n | 3.1 | n | n | n | n | n | n | n | [19] |
| Southern yellow pine | 400 | n | n | n | n | 6.0 | n | n | n | n | 0.03 | 0.29 | n | 0.06 | 0.12 | 0.08 | [68] |
| Spruce wood [y] | 1100–1200 | 29 | 63 | 92 | n | 11.1 | 0.3 | n | 0.2 | n | 0.05 | 0.74 | n | 1.34 | 0.17 | n | [63] |
| Sugarcane bagasse [z] | 343 | n | n | n | 0.11 | 5.8 | n | n | n | n | n | n | n | n | n | n | [48] |
| Sugarcane bagasse [z] | 343 | n | n | n | 0.11 | 6.1 | n | n | n | n | n | n | n | n | n | n | [48] |
| Switchgrass [z] | 1000 | n | n | n | 0.10 | 10.8 | 3.5 | n | 1.3 | 79.0 | 1.20 | 6.60 | n | n | n | n | [69] |
| Wheat straw | 600 | n | n | n | 0.31 | 10.0 | 1.0 | n | 1.0 | 79.3 | n | n | n | n | n | n | [70] |
| Wheat straw | n | n | n | n | 0.24 | 9.5 | 2.5 | n | n | n | 0.003 | 0.10 | 0.002 | 0.004 | 0.0009 | n | [5] |

Note: Production temp: production temperature; CC: container capacity; AS: air space; TP: total porosity; BD: bulk density; EC: electrical conductivity; CEC: cation exchange capacity. Pyrolysis was the biochar production method, unless indicated otherwise. "n" means not available. [z]: Biochar production method was not available. [y]: Biochar was produced from gasification. [x]: Two different sewage sludges were selected from two municipal plants.

## 3. Effects of Biochar on Container Substrates

*3.1. Physical Properties*

### 3.1.1. Bulk Density

The addition of biochar affects the physical properties of container substrates. Biochars have a higher bulk density than commonly-used substrate components, such as peat moss, perlite, and vermiculite. Using biochar to replace certain percentages of peat could thus increase the bulk density of the substrates [5,7,18,71,72].

### 3.1.2. Container Capacity, Air Space and Total Porosity

Biochar incorporation in container substrates may affect container capacity, air space, and total porosity. Particle size distribution of the substrate components is important for determining their physical properties [73]. Due to the differing particle sizes of biochars and substrate components, the effects of biochar incorporation on the physical properties of a container substrate will vary. Container capacity is the maximum percent volume of water a substrate can hold after gravity drainage [74]. Container substrates absorb water in small pores (micropores) between, or inside component particles [10]. Méndez et al. [75] showed that the incorporation of 50% (by vol.) biochar with peat increased container capacity, compared to those with 100% peat substrate due to increased micropores after biochar incorporation. Similar to these results, Zhang et al. [21] also reported that mixing 20% or 35% (w/w) biochar with compost made from green waste increased container capacity. Yet, some research has shown that the incorporation of biochar in container substrates had no effect on container capacity [5,18]. The differing results after biochar incorporation could be due to the different particle sizes of the biochars and the substrate components used. Besides container capacity, biochar incorporation could also affect air space. Air space is the proportion of air-filled large pores (macropores) after drainage [10]. Méndez et al. [75] showed that the incorporation of 50% (by vol.) biochar with peat increased the air space compared to 100% peat substrate. In this study, the percentage of particle size larger than 2 mm was 29% (w/w) for biochar but 8.8% for peat. Thus, the increased air space was caused by an increased number of macropores due to the incorporation of biochar with larger particle size. Zhang et al. [21] confirmed this by showing that mixing biochar with compost increased the percentage of particles larger than 2 mm and thus increased the air space. Total porosity is the sum of air space and container capacity. The effect of biochar on total porosity is related to its effect on air space and container capacity. Substituting peat with 50% biochar (by vol.) made from green waste had no effect on total porosity [18]. Méndez et al. [75] concluded that the addition of biochar produced from deinking sludge increased the total porosity. Zhang et al. [21] also showed that mixing biochar with compost increased the total porosity. Vaughn et al. [5] showed that the effects of biochar on total porosity were mixed and there was no specific trend, when mixing biochar with peat. In summary, biochar incorporation could impact total porosity, air space, and container capacity.

*3.2. Chemical Properties*

### 3.2.1. pH

In general, biochar is effective at increasing the pH of container substrates since the pH of biochars used in most research is neutral to basic [21,53,58,59]. Biochar could buffer acidity due to the negative charge on the surface of biochar [76]. However, the pH of biochars could be acidic. The pH of the biochar depends on the nature of the feedstock and the temperatures during biochar production. The lower the temperature of production, the lower the pH of the biochar. The pH of oak wood biochar was 4.8 when produced at 350 °C [24]. Khodadad et al. [77] also showed pH of biochar made from pyrolysis of oak and grass at 250 °C was 3.5. Lima et al. [78] showed that the pH was around 5.9 for biochars made from pecan shell at 350 °C and switchgrass at 250 °C.

### 3.2.2. Electrical Conductivity

Biochar incorporation could increase container substrate EC due to high EC of the biochar used. The EC of biochar was affected by the biochar functional groups (such as fused-ring aromatic structures and anomeric O-C-O carbons), metal oxide precipitates and binding of metals [24,79]. Hossain et al. [80] also found that as pyrolysis temperature increased, EC of the sludge biochars decreased. When incorporating biochar in container substrate, Vaughn et al. [5] showed that mixing 5%, 10%, and 15% (by vol.) pelletized wheat straw and hardwood biochars with container substrates containing peat moss and vermiculite increased the EC. Tian et al. [18] also found that adding 50% (by vol.) biochar made from green waste to peat moss media significantly increased EC. The increased substrate EC after biochar incorporation could be due to the high pH, large surface area, and charge density of the biochar [70].

### 3.2.3. Cation Exchange Capacity

Biochar incorporation could affect CEC and nutrient availability, which is related to the original properties of biochar itself. Surface functional groups, such as carboxylate, carbonyl and ether are responsible for the CEC of biochar [81]. Different biochars have different chemical functional groups. Vaughn et al. [5] found that some volatile materials were removed and wood cellulosic polymers were carbonized in wood biochar after pyrolysis, while wheat straw biochar was less carbonized and had more chemical functionality, which serves as exchange sites for nutrient absorption. It was shown that CEC was higher in a 25% biochar and 75% peat moss mix (by vol.) than that in 100% peat moss [22]. Some biochars can even provide nutrients to the plants due to the high concentration of certain nutrients in the original feedstocks. Some forms of biochars can serve as a source of P and K, which leads to increased availability of these minerals in container substrates and improved fertility [49,66,82].

### 3.3. Effects on the Microbial Activities

Biochar incorporation may affect microbial activity and biomass in containers. Adding biochar can increase pH, available water content, and influx of nutrients as discussed above, thus stimulating microbial communities and increasing microbial biomass. Warnock et al. [83] also indicated that porous biochar with a high surface area could provide shelter for microorganisms. Saito [84] showed that biochar could serve as a microhabitat for arbuscular mycorrhizal fungi. Higher mycorrhizal colonization and plant growth were shown in mixes of biochar and soil in container experiments [85]. However, only a limited amount of research investigated the effects of biochar on microbial activity or inoculation with mycorrhizae in soilless substrates. Increased mycorrhizal colonization was found in containers containing sand and clay in a ratio of 3:1 (by vol.) with activated biochar (2 g per container) [86]. Inoculation with arbuscular mycorrhizas fungus significantly increased *Pelargonium zonale* plant growth in containers with 0%, 30% or 70% (by vol.) biochar with the rest being peat [87]. Biochars produced at different temperatures may have different surface areas and adsorption abilities [88], which could lead to different levels of nutrient retention and effects on microbial activities.

## 4. Effects of Biochar on Plant Growth in Container Substrates

There is an increasing amount of research on the effects of biochar on container-grown plant growth that shows the potential for biochar to be a replacement for commonly-used soilless container substrate components including peat moss, bark, vermiculite, perlite, coir, etc. Mixing biochar in container substrates may have a positive impact on plant growth due to beneficial effects like improved container physical and chemical properties and enhanced nutrient and water retention, as mentioned above. Tian et al. [18] found that mixing biochar made from green waste with peat (50% each, by vol.) increased total biomass and leaf surface area of *Calathea rotundifolia* cv. Fasciata when compared to that of peat substrates alone, because of improved substrate properties and increased nutrient retention after

biochar incorporation. Replacing 10% (by vol.) of peat with sewage sludge biochar enhanced lettuce (*Lactuca sativa*) biomass production by 184%–270% when compared to 100% peat-based substrate, due to increased N, P and K concentrations and microbial activities [19]. Incorporation of biochar produced from pruning waste at 300 °C (pH = 7.53) and 500 °C (pH = 10.3) into peat substrates at the ratio of 50% and 75% (by vol.) increased lettuce biomass when compared to those in peat alone (pH = 6.14), probably because the increased pH after biochar incorporation was more ideal for many crops [20]. Graber et al. [54] tested the effects of mixing three ratios of citrus wood biochar (1%, 3% or 5%, w/w) with commercial container substrates (a mixture of coconut fiber and tuff at a 7:3 ratio by vol.) on the growth of peppers (*Capsicum annuum*) and tomatoes (*Solanum lycopersicum*). The effects included increased leaf area, shoot dry weight (after detaching the fruits), numbers of flowers and fruit of pepper and increased plant height and leaf size of tomato plants compared to those in commercial container substrates. Graber et al. [54] indicated two possible reasons for the responses, increased beneficial microbial populations or low doses of biochar chemicals stimulating plant growth (hormesis). Mixing 20% or 35% (w/w) biochar made from coir in composted green waste medium increased plant height, root and shoot length, and root fresh and dry weight of *Calathea insignis* when compared to one without any biochar incorporation, effects due to increased water retention, optimized total porosity, aeration porosity, water-holding porosity, nutrients, and microbial activities [21]. Overall, increased plant growth after biochar incorporation could be attributed to increased availability of nutrients and improved water retention, both desirable substrate properties.

However, biochar incorporation may not always improve plant growth. Not all biochars are the same (Table 1). The effects of biochars on container-grown plants are variable (Tables 2–5) depending on multiple factors. There are distinct interactions between biochar and different substrate components. Different biochars, biochar incorporation rate, and other components mixed with biochar can contribute to differing results. Furthermore, individual plant responses to biochar also vary. Across studies of the effects of biochar alone on plant growth, without other factors such as irrigation or fertilization rates, (Tables 2–4), 77.3% reported that some biochar addition to container substrates could promote plant growth, and 50% revealed that plant growth or dry weight was suppressed by some biochar in container substrates. Most studies (69.4%) in Tables 2–5 investigated plant growth in container substrates with biochar for 12 weeks or less than 12 weeks. The length of the experiments in these studies varied from 3 weeks to 7 months. Many mechanisms of biochar-plant interactions are not fully understood.

*4.1. Different Plant Species*

The impact of biochar on plant growth differs by species since different plants have different suitable growth conditions or different tolerance to certain stresses. Mixing potato anaerobic digestate with acidified wood pellet biochar (1:1, by vol.) led to higher fresh and dry weight of tomatoes than a peat: vermiculite control, but led to lower fresh and dry weight of marigold (*Calendula officinalis*) plant [71]. The EC of potato anaerobic digestate is high (7.1 dS m$^{-1}$). The different fresh and dry weight responses of tomato and marigold could be due to the salt tolerances of these two plants [71]. Choi et al. [57] also showed that mixes with 20% pine bark and 80% biochar (by vol.) led to higher chrysanthemum (*Chrysanthemum nankingense*) fresh and dry weight, but lower tomato plant fresh and dry weight when compared to the control. The reduced tomato plant fresh weight and dry weight was because tomato usually requires more nutrients than other plants and biochar can hold or capture nutrients. Furthermore, 80% biochar mixes had no effect on lettuce (*Lactuca sativa*) and basil (*Ocimum basilicum*) fresh and dry weights. Altland and Locke [67] also showed that mixes of 20% (by vol.) gasified rice hull biochar with Sunshine Mix #2 fertilized with 100 mg L$^{-1}$ N using ammonium nitrate and 0.9 kg m$^{-3}$ Micromax caused a smaller *Pelargonium x hortorum* shoot dry weight but increased shoot dry weight of tomato plants when compared to the control (Sunshine Mix #2) fertilized at the rate of 100 mg L$^{-1}$ N with a commercial complete fertilizer with micronutrients.

**Table 2.** Summary of the effects of biochar made from different feedstocks mixed with other substrate components on container-grown plants, with percentage of biochar in container substrates less than 50% (by vol.).

| Plant Species | Non-Biochar Components | Biochar Feedstock | Percentage (%, by vol.) of Biochar and the Effects on Plants' Dry Weight/Growth Index (DW/GI) [z] | | | | | | | | Reference |
|---|---|---|---|---|---|---|---|---|---|---|---|
| | | | 1 | 5 | 10 | 15 | 20 | 25 | 30 | 40 | |
| *Buxus sempervirens* × *Buxus microphylla* | Pine bark and 24 g osmocote 18N–6P–12K | Switchgrass | | | =/n | | | =/n | | | [69] |
| *Calendula offcinalis* | Coir | Forest waste | | | =/n | | | =/n | | | [46] |
| *Chrysanthemum nankingense* | Pine bark | Pine wood | | | | | =/= | | | =/= | [57] |
| *Cucumis melo* | Sunshine commercial growing medium | Standard sugarcane bagasse | | | | | | =/= [y] | | | [48] |
| | | Sugarcane bagasse using a pneumatic transport system | | | | | | =/= [y] | | | |
| *Cucurbita pepo* | Sunshine commercial growing medium | Standard sugarcane bagasse | | | | | | =/= [y] | | | |
| | | Sugarcane bagasse using a pneumatic transport system | | | | | | =/= [y] | | | |
| *Euphorbia* × *lomi* | Peat | Conifer wood | | | | | n/= [y] | | | n/+ [y] | [53] |
| *Euphorbia pulcherrima* | Sunshine Mix #1 | Pine wood | | | | | +/= | | | =/= | [55] |
| *Hydrangea paniculata* | Pine bark and 24 g osmocote 18N–6P–12K | Switchgrass | | | =/n | | | -/n | | | [69] |
| *Lactuca sativa* | Peat | Sewage sludge | | | +/+ [y] | | | | | | [19] |
| *Lactuca sativa* 'Black Seeded Simpson' | Pine bark | Pine wood | | | | | n/= | | | n/= | [57] |
| *Lilium longiflorum* | Sunshine Mix #1 | Pine wood | | | | | =[x]/= [y] | | | =[x]/= [y] | [56] |
| *Ocimum basilicum* 'Genovese' | Pine bark | Pine wood | | | | | =/n | | | =/n | [57] |
| *Ocimum basilicum* | Peat | Softwood from spruce wood | | | | | | | =/n | | [63] |
| | | Harwood from poplar | | | | | | | =/n | | |
| *Ocimum basilicum* | 5% vermicompost (VC) with the rest being Berger BM7 | Mixed hardwood | | | | | =/= | | | =/= | [61] |
| | 10% VC with the rest being Berger BM7 | | | | | | =/= | | | =/= | |
| | 15% VC with the rest being Berger BM7 | | | | | | =/= | | | +/= | |
| | 20% VC with the rest being Berger BM7 | | | | | | =/= | | | +/= | |

**Table 2.** *Cont.*

| Plant Species | Non-Biochar Components | Biochar Feedstock | Percentage (%, by vol.) of Biochar and the Effects on Plants' Dry Weight/Growth Index (DW/GI) [z] | | | | | | | | Reference |
|---|---|---|---|---|---|---|---|---|---|---|---|
| | | | 1 | 5 | 10 | 15 | 20 | 25 | 30 | 40 | |
| *Petunia hybrida* | Coir | Forest waste | | | =/n | | | =/n | | | [46] |
| *Solanum lycopersicum.* 'Red Robin' | 50% vermiculite with the rest being peat and biochar | Pelletized wheat straw | | =/+ [y] | =/+ [y] | =/+ [y] | | | | | [5] |
| | | Hardwood pellets | | =/+ [y] | =/+ [y] | =/+ [y] | | | | | |
| *Solanum lycopersicum* 'Cuarenteno' | Coir | Forest waste | | | =/n | | | =/n | | | [47] |
| | | Olive mill waste | | | +/n | | | =/n | | | |
| *Solanum lycopersicum* 'Gransol Rijk Zwaan' | | Forest waste | | | =/n | | | -/n | | | |
| | | Olive mill waste | | | =/n | | | =/n | | | |
| *Solanum lycopersicum* 'Hope' | Pine bark | Pine wood | | | | | =/= | | | +/= | [57] |
| *Solanum lycopersicum* 'Roma' | 5% VC with the rest being Berger BM7 | Mixed hardwood | | | | | =/= | | | =/= | [61] |
| | 10% VC with the rest being Berger BM7 | | | | | | +/= | | | =/= | |
| | 15% VC with the rest being Berger BM7 | | | | | | =/+ | | | +/= | |
| | 20% VC with the rest being Berger BM7 | | | | | | =/= | | | =/= | |
| *Tagetes erecta* 'Inca II Yellow Hybrid' | 50% vermiculite with the rest being peat and biochar | Pelletized wheat straw | | =/+ [y] | =/+ [y] | =/+ [y] | | | | | [5] |
| | | Hardwood pellets | | =/= [y] | =/+ [y] | =/+ [y] | | | | | |

[z]: "+" means increased; "=" means there was no significant difference; "-" means decreased; "n" means not available. [y]: Result for this was for plant height not growth index. [x]: Result for this was for leaf dry height not total dry weight.

**Table 3.** Summary of the effects of biochar made from different feedstocks mixed with other substrate components on container-grown plants, with percentage of biochar in container substrates ranging from 50% to 100% (by vol.).

| Plant Species | Non-Biochar Components | Biochar Feedstock | Percentage (%, by vol.) of Biochar and Its Effect on Plants' Dry Weight/Growth Index (DW/GI) [z] | | | | | | | | Reference |
|---|---|---|---|---|---|---|---|---|---|---|---|
| | | | 50 | 60 | 66 | 70 | 75 | 80 | 90 | 100 | |
| *Anethum graveolens* | Perlite | Rice husk | +/+ [y] | | | | | | | | [64] [w] |
| *Brassica rapa* ssp. *pekinensis* | | | +/+ [y] | | | | | | | | |
| *Brassica rapa* var. *rosularis* | | | +/+ [y] | | | | | | | | |

**Table 3.** *Cont.*

| Plant Species | Non-Biochar Components | Biochar Feedstock | Percentage (%, by vol.) of Biochar and Its Effect on Plants' Dry Weight/Growth Index (DW/GI) z | | | | | | | | Reference |
|---|---|---|---|---|---|---|---|---|---|---|---|
| | | | 50 | 60 | 66 | 70 | 75 | 80 | 90 | 100 | |
| *Calathea rotundifolia* cv. Fasciata | Peat | Green waste | +/n | | | | | | | | [18] |
| *Chrysanthemum nankingense* | Pine bark | Pine wood | | =/n | | | | +/n | | =/n | [57] |
| *Cucumis melo* | Sunshine commercial growing medium | Standard sugarcane bagasse | =/= y | | | | -/= y | | | -/= y | [48] |
| | | Sugarcane bagasse using a pneumatic transport system | =/+ y | | | | =/= y | | | -/= y | |
| *Cucurbita pepo* | Sunshine commercial growing medium | Standard sugarcane bagasse | =/= y | | | | =/= y | | | =/= y | |
| | | Sugarcane bagasse using a pneumatic transport system | +/+ y | | | | =/= y | | | =/= y | |
| *Euphorbia* × *lomi* | Peat | Conifer wood | | +/+ y | | | | +/+ y | | n/= y | [53] |
| *Euphorbia pulcherrima* | Sunshine Mix #1 | Pine wood | | -/= | | | | -/= | | -/- | [55] |
| *Lactuca sativa* | Perlite | Rice husk | +/+ y | | | | | | | | [64] w |
| *Lactuca sativa* | Peat | Deinking sludge | +/n | | | | | | | | [75] |
| | Coir | | -/n | | | | | | | | |
| *Lactuca sativa* | Peat | Pruning wastes | +/n | | | | +/n | | | | [20] |
| *Lactuca sativa* 'Black Seeded Simpson' | Pine bark | Pine wood | | n/= | | | | n/= | | n/= | [57] |
| *Lilium longiflorum* | Sunshine Mix #1 | Pinewood | | = x/= y | | | | = x/= y | | | [56] |
| *Malva verticillata* | Perlite | Rice husk | +/+ y | | | | | | | | [64] w |
| *Ocimum basilicum* | 5% VC with the rest being Berger BM7 | Mixed hardwood | | =/= | | | | | | | [61] |
| | 10% VC with the rest being Berger BM7 | | | =/= | | | | | | | |
| *Ocimum basilicum* | 15% VC with the rest being Berger BM7 | Mixed hardwood | | +/= | | | | | | | [61] |
| | 20% VC with the rest being Berger BM7 | | | =/= | | | | | | | |
| *Ocimum basilicum* | 5% chicken manure compost (CM) with the rest being Berger BM7 | Mixed hardwood | | =/= | | =/= | -/- | | | | [61] |
| *Ocimum basilicum* | 5% VC with the rest being Berger BM7 | | | =/= | | =/= | -/- | | | | |

**Table 3.** *Cont.*

| Plant Species | Non-Biochar Components | Biochar Feedstock | Percentage (%, by vol.) of Biochar and Its Effect on Plants' Dry Weight/Growth Index (DW/GI) [z] | | | | | | | | Reference |
|---|---|---|---|---|---|---|---|---|---|---|---|
| | | | 50 | 60 | 66 | 70 | 75 | 80 | 90 | 100 | |
| *Ocimum basilicum* 'Genovese' | Pine bark | Pine wood | | =/n | | | | =/n | | -/n | [57] |
| *Solanum lycopersicum* 'Red Robin' | Potato digestate | Wood pellet | +/=[y] | | | | | | | | [71] |
| | | Pelletized wheat straw | +/=[y] | | | | | | | | |
| | | Pennycress presscake | =/=[y] | | | | | | | | |
| *Solanum lycopersicum* 'Gransol Rijk Zwaan' | Coir | Forest waste | -/n | | | | -/n | | | -/n | [47] |
| | | Olive mill waste | -/n | | | | -/n | | | -/n | |
| *Solanum lycopersicum* 'Cuarenteno' | Coir | Forest waste | -/n | | | | -/n | | | -/n | |
| | | Olive mill waste | =/n | | | | =/n | | | =/n | |
| *Solanum lycopersicum* | Faecal sludge based compost | Rice husk | =/=[y] | | | | | | | -/-[y] | [65] |
| *Solanum lycopersicum* 'Roma' | 5% VC with the rest being Berger BM7 | Mixed hardwood | | =/= | | | | | | | [61] |
| | 10% VC with the rest being Berger BM7 | | | =/= | | | | | | | |
| | 15% VC with the rest being Berger BM7 | | | =/= | | | | | | | |
| | 20% VC with the rest being Berger BM7 | | | =/= | | | | | | | |
| *Solanum lycopersicum* 'Tumbling Tom Red" | 5% CM with the rest being Berger BM7 | Mixed hardwood | | +/= | | =/= | -/- | | | | [61] |
| | 5% VC with the rest being Berger BM7 | | | =/= | | =/= | =/= | | | | |
| *Solanum lycopersicum* 'Hope' | Pine bark | Pine wood | | =/= | | | | -/= | | -/= | [57] |
| *Tagetes erecta* | Potato digestate | Wood pellet | -/=[y] | | | | | | | | [71] |
| | | Pelletized wheat straw | =/=[y] | | | | | | | | |
| | | Pennycress presscake | -/=[y] | | | | | | | | |

[z]: "+" means increased; "=" means there was no significant difference; "-" means decreased; "n" means not available. [y]: Result for this was for plant height not growth index. [x]: Result for this was for leaf dry height not total dry weight. [w]: Hydroponic experiment.

**Table 4.** Summary of the effects of biochar made from different feedstocks mixed with other substrate components on container-grown plants, with percentage of biochar in container substrates measured by weight.

| Plant Species | Non-Biochar Components | Biochar Feedstock | Percentage (%, by weight) of Biochar and Its Effect on Plants' Dry Weight/Growth Index (DW/GI) [z] | | | | | | | | | | Reference |
|---|---|---|---|---|---|---|---|---|---|---|---|---|---|
| | | | 1 | 2.5 | 3 | 5 | 10 | 20 | 35 | 40 | 60 | 80 | |
| *Acmena smithii* | Growing medium (pine bark, coir, clinker ash and coarse sand) with 3 kg m$^{-3}$ controlled-release fertilizer (CRF) | *Eucalyptus saligna* wood chip | | =/n | | =/n | =/n | | | | | | [51] |
| | Growing medium (pine bark, coir, clinker ash and coarse sand) with 6 kg m$^{-3}$ CRF | | | +/n | | =/n | =/n | | | | | | |
| | Growing medium (pine bark, coir, clinker ash and coarse sand) with no CRF | | | +/n | | =/n | =/n | | | | | | |
| *Calathea insignis* | Composted green waste medium | Coir (coconut husk fiber) | | | | | | +/+ [y] | +/+ [y] | | | | [21] |
| | 0.5% humic acid (w/w) with the rest being green waste compost | | | | | | | +/+ [y] | +/+ [y] | | | | |
| | 0.7% humic acid (w/w) with the rest being green waste compost | | | | | | | +/+ [y] | +/= [y] | | | | |
| *Capsicum annuum* | A mixture of coconut fiber and tuff at a ratio of 7:3 (by vol.) | Citrus wood | n/= [y] | | n/= [y] | | | | | | | | [54] |
| *Capsicum annuum* | Sphagnum peatmoss-based medium in 50-cell transplant trays | Hardwood including oak, elm, and hickory | | | | | | =/= [y] | | =/= [y] | -/- [y] | -/- [y] | [60] |
| | Sphagnum peatmoss-based medium in 72-cell transplant trays | | | | | | | =/+ [y] | | =/= [y] | =/= [y] | -/- [y] | |
| | Sphagnum peatmoss-based medium in 98-cell transplant trays | | | | | | | =/= [y] | | =/= [y] | =/= [y] | -/- [y] | |
| *Solanum lycopersicum* | A mixture of coconut fiber and tuff at a ratio of 7:3 (by vol.) | Citrus wood | n/+ [y] | | n/+ [y] | | | | | | | | [54] |
| *Viola × hybrida* | Growing medium (pine bark, coir, clinker ash and coarse sand) blended with 3 kg m$^{-3}$ CRF | *Eucalyptus saligna* wood chip | | =/n | | =/n | =/n | | | | | | [51] |
| | Growing medium (pine bark, coir, clinker ash and coarse sand) blended with 6 kg m$^{-3}$ CRF | | | +/n | | =/n | +/n | | | | | | |
| | Growing medium (pine bark, coir, clinker ash and coarse sand) with no CRF | | | -/n | | =/n | -/n | | | | | | |
| *Viola × wittrockiana* | Growing medium (pine bark, coir, clinker ash and coarse sand) blended with 3 kg m$^{-3}$ CRF | *Eucalyptus saligna* wood chip | | -/n | | =/n | =/n | | | | | | [51] |
| | Growing medium (pine bark, coir, clinker ash and coarse sand) blended with 6 kg m$^{-3}$ CRF | | | +/n | | =/n | =/n | | | | | | |
| | Growing medium (pine bark, coir, clinker ash and coarse sand) with no CRF | | | +/n | | =/n | =/n | | | | | | |

[z]: "+" means increased; "=" means there was no significant difference; "-" means decreased; "n" means not available. [y]: Result for this was for plant height not growth index.

**Table 5.** Other studies testing the effects of biochar mixed with other substrate components on container-grown plants.

| Plant Species | Non-Biochar Components | Biochar Feedstock | Biochar Percentage (by vol.) | Effects on Plant Growth [z] | Other Information | Reference |
|---|---|---|---|---|---|---|
| *Agrostis stolonifera* | Sand | Southern yellow pine | 15% | Plant height (=)/DW (=)/FW (=) | Control was mixes with 85% sand and 15% peat | [68] |
| | 85% sand and 10% peat, vermicompost, yard-waste compost, Organimix compost, humus or worm castings | | 5% | Plant height (=)/DW (=)/FW (=) | | |
| | 85% sand and 10% anaerobic biosolids | | 5% | Plant height (+)/DW (+)/FW (+) | | |
| *Helianthus annuus* | Pig slurry compost | Holm oak | 40% or 80% | Shoot DW (+) | Compared to mixes with 40% or 80% coir with the rest being pig slurry compost, respectively | [52] |
| | Pig slurry compost | | 60% | Shoot DW (=) | Control was mixes with 60% coir with the rest being pig slurry compost | |
| | No | | 100% | Shoot DW (=) | Control was 100% coir | |
| *Ipomoea aquatica* | Spent pig litter compost, vermiculite, perlite and peat | Wheat straw | 2%, 4% or 8% | Germination rate (=) | | [70] |
| | | | 10%, 12%, 14% or 16% | Germination rate (-) | | |
| *Lactuca sativa* | Two parts of single wood species sawdust to one-part poultry manure | Rice husk | 50% or 66% | Plant height (-) | Half irrigation (0.1125mm) | [92] |
| | | | 50% | Plant height (=) | Full irrigation (0.225mm) | |
| | | | 66% | Plant height (+) | Full irrigation (0.225mm) | |
| *Pelargonium* × *hortorum* 'Maverick Red' | Sunshine Mix #2 | Rice hull | 1% or 10% | Shoot DW (-) | Plants in biochar-added substrates were fertilized with 100 mg $L^{-1}$ N. Control was Sunshine Mix #2 with a fertilizer (20N-4.4P-16.6K-0.15Mg-0.02B-0.01Cu-0.1Fe-0.05Mn-0.01Mo-0.05Zn) at the rate of 100 mg $L^{-1}$ N | [66] |
| | Sunshine Mix #2 with a micronutrient package (Micromax, The Scotts Co., Marysville, OH) at 0.9 kg $m^{-3}$ | Rice hull | 5%,10% or 15% | Shoot DW (=) | | [67] |
| | | | 20% | Shoot DW (-) | | |
| *Pelargonium zonale* | Peat | N/A | 30% or 70% | Plant height (=)/DW (=) | 140 mg $L^{-1}$ slow released fertilizer applied | [87] |
| | | | 30% | Plant height (=)/DW (=) | 210 mg $L^{-1}$ slow released fertilizer applied | |
| | | | 70% | Plant height (-)/DW (-) | | |
| *Solanum lycopersicum* 'Megabite' | Sunshine Mix #2 with a micronutrient package (Micromax, The Scotts Co., Marysville, OH) at 0.9 kg $m^{-3}$ | Rice hull | 5% | Shoot DW (=) | Plants in biochar-added substrates were fertilized with 100 mg $L^{-1}$ N. Control was Sunshine Mix #2 with a fertilizer (20N-4.4P- 16.6K-0.15Mg-0.02B-0.01Cu-0.1Fe-0.05Mn-0.01Mo-0.05Zn) at the rate of 100 mg $L^{-1}$ N | [67] |
| | | | 10%, 15% or 20% | Shoot DW (+) | | |
| *Solanum lycopersicum* | Pine (*Pinus radiata* D. Don) sawdust | Tomato crop green waste | 25%, 50%, 75% or 100% | Shoot fresh weight (FW) (=)/Fruit number (=)/ Yield (=) | Control was 100% pine sawdust. | [59] |

**Table 5.** *Cont.*

| Plant Species | Non-Biochar Components | Biochar Feedstock | Biochar Percentage (by vol.) | Effects on Plant Growth [z] | Other Information | Reference |
|---|---|---|---|---|---|---|
| *Sylibum marianum* | Pig slurry compost | Holm oak | 40%, 60% or 80% | Shoot DW (=) | Compared to mixes with 40%, 60%, or 80% coir with the rest being pig slurry compost, respectively | [52] |
| | No | | 100% | Shoot DW (-) | Control was 100% coir | |
| *Tagetes erecta* | 30% perlite with the rest being peat and biochar | Mixed softwood | 10%, 20%, 30%, 40%, 50%, 60% or 70% | Plant height (=)/DW (=) | No pH adjustment; control was 70:30 peat: perlite mixture. | [62] |
| | | | 10% or 70% | Plant height (-)/DW (=) | pH adjusted to 5.8; control was 70:30 peat: perlite mixture. | |
| | | | 20%, 30%, 40%, 50% or 60% | Plant height (=)/DW (=) | | |
| *Zelkova serrata* | Growing medium mixture of peat moss, perlite, and vermiculite at a ratio of 1:1:1 by vol. | Pine chip | 20% | Plant height (=)/Stem DW (=) | 0.5 or 1 g/L fertilization | [50] |
| | | Oak chip | 20% | Plant height (=)/Stem DW (=) | 0.5 or 1 g/L fertilization | |
| | | Pine cone | 20% | Plant height (-)/Stem DW (-) | 0.5 g/L fertilization | |
| | | | | Plant height (=)/Stem DW (=) | 1 g/L fertilization | |
| | | Rice husk | 20% | Plant height (+)/Stem DW (+) | 0.5 g/L fertilization | |
| | | | | Plant height (=)/Stem DW (=) | 1 g/L fertilization | |
| | | Crab shell | 20% | Plant height (-)/Stem DW (-) | 0.5 or 1 g/L fertilization | |

[z]: "+" means increased; "=" means there was no significant difference; "-" means decreased; "n" means not available.

(20N-4.4P-16.6K-0.15Mg-0.02B-0.01Cu-0.1Fe-0.05Mn-0.01Mo-0.05Zn). The effects of biochar on container-grown plants could be different due to different plant materials.

Most of the plant species used in testing biochars in container substrates have been herbaceous. Only six woody plants have been tested, including Japanese zelkova (*Zelkova serrata*), lilly pilly (*Acmena smithii*), 'Green Velvet' boxwood (*Buxus sempervirens* × *Buxus microphylla*), Pinky Winky hardy hydrangea (*Hydrangea paniculata*), myrtle (*Myrtus communis*) and mastic tree (*Pistacia lentiscus*). Across all studies, the most frequently tested species have been tomato and lettuce. About 30.5% of the studies used tomato plants to test biochars in container substrates and 19.4% used lettuce. Research is needed to test more plant species.

## 4.2. Different Biochar and Biochar Percentage in Container Substrates

The impact of biochar on plant growth depends on the properties of the biochar used and the percentage of biochar in the substrates. Those factors impact the overall physical and chemical properties of the container substrates, such as pH, container capacity and CEC. Belda et al. [89] showed that mixing 10%, 25% or 50% (by vol.) forest waste biochar with coir led to higher *Myrtus communis* and *Pistacia lentiscus* stem length and dry weight than using olive mill waste biochar. It was shown that *Zelkova serrata* plants in mixes that contained 20% rice husk biochar with the rest of the mixture composed of peat moss, perlite, and vermiculite at a ratio of 1:1:1 (by vol.) were 6 times larger than those in mixes with crab shell biochar, which could be due to the high concentration of nutrients, nutrient absorption ability and water retention ability of rice husk biochar [50]. Webber et al. [48] showed that pneumatic sugarcane bagasse biochar and standard sugarcane bagasse biochar led to different effects on plant growth, due to different physical and chemical compositions of the two biochars, produced by different conditions. Pumpkin (*Cucurbita pepo*) and muskmelon (*Cucumis melo*) both had increased plant height in mixes with 50% pneumatic sugarcane bagasse biochar with the rest being Sunshine commercial growing media (by vol.) compared to the control, while both in mixes with 50% standard sugarcane bagasse biochar showed similar plant height to the control. Webber et al. [48] also indicated that different biochar percentages could affect the results and showed that mixes with 75% or 100% biochar decreased muskmelon plant dry weight, but mixes with 25% or 50% biochar had no effect. Similarly, the aboveground dry weight of *Viola* × *hybrida* showed no significant effects after the incorporation of 5% (w/w) *Eucalyptus saligna* wood chip biochar to growing medium containing pine bark, coir, clinker ash and coarse sand, but aboveground dry weight decreased when mixing 10% (w/w) biochar with the growing medium, when compared to the control [51]. The decreased plant dry weight was due to reduced concentrations of S, P, and Ca caused by the binding ability of the biochar [51]. Fan et al. [70] found that the germination rate of water spinach (*Ipomoea aquatica*) decreased when the biochar incorporation rate in mixes containing spent pig litter compost, vermiculite, perlite and peat increased to 10%, 12%, 14% or 16% (by vol.) due to the high and unsuitable pH and EC after biochar incorporation, while there was no effect on the germination rate if the biochar incorporation rate was 2%, 4% or 8% (by vol.). Conversa et al. [87] showed that mixing peat with biochar at the ratio of 70:30 (by vol.) with slow released fertilizer at a rate of 140 and 210 mg L$^{-1}$ led to increased *Pelargonium* leaf number and similar shoot dry weight compared to the control. However, mixing peat with biochar at the ratio of 30:70 (by vol.) with a high rate of slow release fertilizer (210 mg L$^{-1}$) showed decreased *Pelargonium* plant growth and flowering traits due to osmotic stress caused by high EC and decreased mycorrhizal activity with this high biochar rate [87]. Awad et al. [64] also showed that mixes with 50% (by vol.) biochar with the rest being perlite led to increased dry weight and growth of Chinese cabbage (*Brassica rapa* ssp. *pekinensis*), dill (*Anethum graveolens*), curled mallow (*Malva verticillata*), red lettuce, and tatsoi (*Brassica rapa* var. *rosularis*) while 100% rice husk biochar decreased plant growth due to high pH of the substrate, low air space, and decreased N availability due to biochar's N absorption ability.

Across studies that mixed biochar in container substrates by volume and tested the effects of biochar on plant growth without other factors (Tables 2 and 3), 72.2% incorporated biochar at 50% or

more (by vol.) in container substrates. This suggested that the substitution of the commonly-used substrates or substrate components with a large proportion of biochar is highly desired and, based on the results, achievable. About 36.4% of the studies (Tables 2 and 3) showed that mixing high percentages of biochar (at least 50% by vol.) in container media could improve the growth of some species when compared to the control. All container substrates with biochar percentages lower than 25% (by vol.) led to similar or higher plant growth or dry weight when compared to the control. A biochar incorporation rate as high as 100% (by vol.) in container substrate often led to similar plant growth to the control [48,53,57].

The physical and chemical properties of biochar could determine whether a large proportion of biochar could be used in container substrates to grow plants. When the physical and chemical properties of biochar or substrates with high percentages of biochar are similar to the commercial substrates or are in the ideal range for container-grown plant growth, a high percentage of biochar could be incorporated into the container substrate. The recommended ranges for the physical properties of most substrates used in commercial container plant production are 50%–85% for total porosity, 10%–30% for air space, 45%–65% for container capacity and 0.19 to 0.7 g cm$^{-3}$ for bulk density [72]. Choi et al. [57] has achieved using 100% biochar substrates to replace the 100% pine bark substrates to grow chrysanthemum and lettuce. The container capacity and air space of the biochar were similar to the bark [57]. Although the total porosity of the biochar used was different from that of the bark, it was in the recommended range for container plant production [57]. Guo et al. [56] also succeeded using up to 80% biochar in peat-based commercial substrates, and the physical properties of the biochar substrates were in, or close to, the recommended range for container plant production. Among all properties, pH could be a limiting factor determining the potential use of biochar in containers. Webber et al. [48] made two kinds of biochars, pneumatic sugarcane bagasse biochar and standard sugarcane bagasse biochar, and indicated that these two biochars could be used in containers as high as 100% to grow pumpkin seedlings for 20 days. The pH of these two bicohars were 5.8 and 6.05, respectively. If the pH of the biochar is high, other acidic components should be added to reduce the pH or a high percentage of biochar in a container may not be achievable. It was shown that the addition of 80% (by vol.) biochar (pH = 8.5) to peat (pH = 5.7) increased plant growth due to neutral pH and improved water holding and air structure after biochar addition [53].

### 4.3. Other Substrate Components Mixed with Biochar in Container Substrates

The other substrate components used with biochar could affect plant growth due to their different physical and chemical properties and their effects on the overall container substrate properties. Substrate components mixed with biochar have included peat, vermiculite, perlite, coir, pine bark, pine sawdust, commercial growing media, compost, composted green waste and potato digestate (Tables 2–5). Gu et al. [90] showed that gomphrena (*Gomphrena globosa*) grown in 5%, 10%, 15%, 20%, 25% and 30% (by vol.) pinewood biochar mixed with the peat-based Sunshine Mix #1 had greater width and height, higher fresh weight and dry weight than those grown in biochar mixed with bark substrates at 43 days after transplanting. The reason for this result could be that peat-based substrates have more organic matter and higher water and nutrient holding capacity than bark-based substrates. Ain Najwa et al. [91] also indicated that the fruit number and fresh weight of tomato in mixes with coco peat and 150 g biochar were higher than in mixes with oil palm fruit bunch (a newly developed organic medium) and 150 g biochar due to different physical and chemical properties of these two substrates. Vaughn et al. [68] showed that creeping bentgrass (*Agrostis stolonifera*) had higher fresh and dry weight and shoot height in mixes with 85% sand, 10% anaerobic biosolids and 5% biochar (by vol.) than the one in mixes with 85% sand, 10% peat and 5% biochar (vol.), due to higher nitrate concentration caused by biosolid incorporation. Méndez et al. [75] also demonstrated that the total biomass and shoot and root weight of lettuce were higher in deinking sludge biochar with peat (50:50 by vol.) than those in biochar mixed with coir (50:50 by vol.). The lower plant biomass in coir with biochar incorporation may be due to the lower CEC, N and P in coir when compared to peat. Fan et al. [70]

investigated the effects of mixed wheat straw biochar with or without superabsorbent polymer on the substrates containing spent pig litter compost, vermiculite, perlite and peat. The germination rate of water spinach decreased when the biochar incorporation rate in the medium without superabsorbent polymer was 10%, 12%, 14% or 16% (by vol.) due to the high and unsuitable pH and EC after biochar incorporation. However, there was no difference on germination rate between the mixes with different percentages of biochar (from 0% to 16% by vol.) when biochar was applied together with superabsorbent polymer. The reason was that the incorporation of superabsorbent polymer increased the porosity and water-holding capacity and also effectively prevented an excessive increase of pH and EC at the high biochar rates [70]. Margenot et al. [62] also showed that mixes with 10%, 20%, 30%, 40%, 50%, 60% or 70% softwood biochar and 30% perlite with the rest being peat (by vol.) led to similar seed germination and plant height compared to control (mixes with 30% perlite and 70% peat by vol.). However, if other components such as calcium hydroxide were added to increase the pH of 10% biochar mixes to 5.8 or pyroligneous acid to decrease substrate (mixes with more than 10% biochar) pH, lower seed germination resulted in mixes with 50%, 60% or 70% biochar and lower plant height in mixes with 10% or 70% biochar.

## 5. Effect of Potentially Toxic Contaminants in Biochar on Plant Growth

Biochar may contain potentially toxic substances, such as heavy metals and organic contaminants (PAH and dioxin), which are affected by the production conditions and feedstocks used. The incorporation of biochar with a high content of these contaminants is a concern. Various studies have shown reduced plant growth caused by the toxicity of PAHs [93,94], dioxins [95] and heavy metals [96,97]. The utilization of biochar that contains toxic substances could be detrimental, and could influence plant growth and development, leach into groundwater, and have noxious effects on soil function and microorganisms. However, toxic substances (heavy metals, PAHs and dioxin) in biochars used in container substrates have rarely been tested. Attention is needed when choosing biochar feedstocks and biochar production conditions to avoid or minimize the production of toxic substances.

Biochar could contain heavy metals from contaminated feedstocks; however, heavy metals could be transformed to more stable forms after pyrolysis, thus having less effect on plant growth. Heavy metals may remain in biochar made from contaminated feedstock such as cadmium (Cd), copper (Cu), lead (Pb), and zinc (Zn) as observed with contaminated willow leaves and branches [98] or sewage [99]. However, the heavy metals in biochar might have low bioavailability after pyrolysis and a lower risk to plant growth. Jin et al. [100] found most of the heavy metals in sludge biochar after pyrolysis at 400 to 600 °C, including Cu, Zn, Pb, chromium (Cr), manganese (Mn) and nickel (Ni), were in their oxidized and residual forms, which had low bioavailability and thus risks. Similarly, Devi and Saroha [101] found that the bioavailability of heavy metals (Cr, Cu, Ni, Zn and Pb) in paper mill sludge biochar derived from pyrolysis at 200 °C to 700 °C was reduced due to transformation into more stable forms. Buss et al. [102] investigated the effects of 19 types of biochar produced from marginal biomass containing contaminants (such as Cu, Cr, Ni and Zn) on plant growth and found that only five types of biochar in the study showed suppressive effects on plant growth after adding 5% (by weight) of biochar in sand due to high K and pH, not heavy metals.

Although PAHs could be formed in biochars due to production conditions, the amount of PAH in biochars used in many studies has been low and may have had low toxicity for plant growth. Large quantities of PAHs are formed in reactions at high temperatures, especially over 750 °C [103], although no research was found using biochar produced over 750 °C in container substrates. There is also evidence that small amounts of PAHs can be formed in pyrolysis reactors operating between 400 °C and 600 °C [103,104], which is the temperature range that most biochars suitable as container substrate component were produced [6,19–21,51,70,75,90]. Research has shown that PAHs in biochar produced from slow pyrolysis between temperature 250 °C and 900 °C had very low bioavailability [105]. Wiedner et al. [29] also found that all biochars made from gasification of poplar, wheat straw, sorghum and olive, and from pyrolysis of draff (the waste product from the production of beer after separating

liquid malt) and miscanthus contained very low content of PAH (below 1.7 mg kg$^{-1}$) and biochar made from woodchip gasification (15 mg kg$^{-1}$ PAH). Although biochars produced at certain conditions, especially over 750 °C, could contain PAHs, no research was found using these biochars in container substrates to test their effects on substrate properties and plant growth.

Dioxins could be formed in biochar if the feedstock contains chlorine in certain conditions, but dioxin concentration in biochars could be very low and have a negligible effect on plant growth. Dioxins refer to compounds such as polychlorinated dibenzo dioxins (PCDDs) and polychlorinated dibenzo furans (PCDFs), which are persistent organic pollutants [106]. Dioxins could be formed only in biochars made from feedstock containing chlorine, such as straws, grasses, halogenated plastics and food waste containing sodium chloride under specific conditions [103,106]. Dioxins could be produced during two pathways: "precursor" pathway, which begins with the synthesis of dioxin precursors from feedstock containing chlorine at temperatures between 300 °C and 600 °C; and the "de novo" pathway, which occurs between 200 °C and 400 °C in a catalytic reaction with oxygen and carbon [106–108]. However, the dioxin in biochar made from feedstock with chlorine could be very low. Hale et al. [105] investigated the biochars produced at 250 °C to 900 °C via slow pyrolysis, fast pyrolysis and gasification and found that total dioxin concentrations in biochars tested were very low (92 pg g$^{-1}$) and bioavailable concentrations were below detection limit [105]. Wiedner et al. [29] found that the dioxins in four biochars produced from gasification of poplar and olive residues and pyrolysis of draff and wood chips and two other hydrochars made from leftover food and sewage sludge were all under the limit of detection, except the one made from sewage sludge (14.2 ng kg$^{-1}$). No evidence was found testing the effect of biochars with dioxin in container substrates on plant growth.

## 6. Discussion

The incorporation of biochar into container substrates could affect physical and chemical properties of the container substrates and thus contribute to the growth of container-grown plants. Most biochars have a higher bulk density than commonly-used substrates, and thus the incorporation of biochar could increase the bulk density of the container substrate. The effect of biochar on container capacity, air space, and total porosity of the container substrates depends on the particle size distribution of the biochar and the other components in the container. The liming effect of alkaline biochars could adjust the container substrate with low pH to an optimal pH. In addition, biochar incorporation could increase EC, nutrient availability, and CEC.

The effects of biochar on plant growth in container substrates varies as not all biochars are the same. The characteristics of biochars differ according to the feedstock used and the pyrolysis process. Many factors, such as plant species and the ratio of biochar to other container substrate components, can contribute to different results on container substrate properties and plant growth. Across studies testing the effects of biochar on plant growth but not other factors (such as irrigation or fertilization rates) (Tables 2–4), 77.3% of the studies found that plant growth could be increased by the incorporation of certain percentages of biochar in container substrates, and 50% revealed that certain percentages of biochar addition could decrease plant growth. Among studies mixing biochar with container substrates by volume and testing the effects of biochar on plant growth without other factors (Tables 2 and 3), 36.4% showed that container substrates with high percentages of biochar (at least 50% by vol.) could improve plant growth under certain conditions compared to the control. All the container substrates with biochar percentages lower than 25% (by vol.) led to similar or higher plant growth or dry weight when compared to the control. A biochar incorporation rate as high as 100% (by vol.) in container substrates could lead to similar plant growth to the control. The physical and chemical properties of the biochar could determine whether a large proportion of biochar could be used in container substrates to grow plants.

There is no universal standard for using biochar in container substrates for all plants. Many mechanisms of biochar are not fully understood. Research on biochar in container substrates is still in an exploratory state. Most research has focused on testing whether biochar could be used

to substitute for commonly-used substrates such as peat, perlite and bark in containers to grow plants, and compared plant growth with a control that had no biochar addition. There is very limited research that tests other properties such as the effect of biochar on disease suppression in container substrates. Research has shown that biochar could impact greenhouse gas emissions in soil, but limited research has been conducted on soilless container substrates. A limited number of published studies have investigated the effect of biochar on microbial activity or inoculation with mycorrhizae in containers. Most of the species used in reported studies testing biochar in container substrates have been herbaceous plants. More plant species should be used to test the effects of biochar to broaden its use. Future studies could be focused on biochars with promising results, to fine-tune the pyrolysis process and incorporate formulae for diverse container substrates.

**Author Contributions:** This review is a product of the combined effort of both authors. L.H. wrote the original draft and improved it based on M.G.'s advice and assistance. M.G. reviewed, edited and revised the manuscript.

**Conflicts of Interest:** The authors declare no conflict of interest.

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
