# Peer review of "Effects of Biochar on Container Substrate Properties and Growth of Plants—A Review"

_horticulturae, doi:10.3390/horticulturae5010014_

Round 1

Reviewer 1 Report

I thank the authors for the revision of the manuscript. I feel that despite the great effort of authors gathering such large amount of information, the manuscript is still characterized by large descriptive sections, without a critical evaluation of the results in order to highlight the most important finding and research gaps needed. An example is the use of extremely large table with lots of information that can not be easily scrutinized. The section of biochar production and presence of pollutants are interesting but I feel they are descriptive without a clear focus on the aim of the review (how production affects properties – with data; and how the presence of pollutants limit plant growth – critical evaluation). For this reason I am afraid I keep my original opinion on the mansucript.

Author Response

Reviewer #1:

Comments and Suggestions: I feel that despite the great effort of authors gathering such large amount of information, the manuscript is still characterized by large descriptive sections, without a critical evaluation of the results in order to highlight the most important finding and research gaps needed. An example is the use of extremely large table with lots of information that cannot be easily scrutinized. The section of biochar production and presence of pollutants are interesting but I feel they are descriptive without a clear focus on the aim of the review (how production affects properties – with data; and how the presence of pollutants limit plant growth – critical evaluation).

We have improved our manuscript a lot by adding more quantitative results and critical evaluation of the results. Research gaps were also mentioned in the manuscript. The two tables in our manuscript were closely related to our main topic and we believe that two tables could provide references to the other researchers when testing biochar in container substrates on plant growth. For section 5 about the potentially toxic contaminants in biochar, we would like to provide this information to the readers and remind them to be more careful about choosing biochar feedstock and biochar production conditions. Toxic substances (heavy metals, polycyclic aromatic hydrocarbons and dioxin) in biochars used in container substrates were rarely tested in research.

Reviewer 2 Report

The content of this review article on biochar use in container substrates is thorough and comprehensive.  I think this paper would be a very useful addition to the literature.  However, the paper needs a lot of work to correct grammar and writing style.  

Author Response

Reviewer #2:

Comments and Suggestions: The content of this review article on biochar use in container substrates is thorough and comprehensive.  I think this paper would be a very useful addition to the literature.  However, the paper needs a lot of work to correct grammar and writing style. 

A native English-speaking colleague has checked our manuscript and all the English grammar and style revisions made by him have been highlighted and incorporated.

Reviewer 3 Report

A commendable effort on a large topic. The continued interest in biochar does necessitate a thorough review of what is known about these materials and the different processes related to them. I do have some concerns that I would like to address and maybe some suggestions.

Overall this review does not "flow well". I found it very choppy and difficult to read. Topics jumped around and the story was buried a bit. A lot of facts presented but the organization may could be improved. 

Line 11: interest of whom?

Introduction: I would delete the first paragraph (lines 29-42). This is a lot of references and space dedicated to biochar in soils, not in soilless substrates. I do not think it is necessary to overview biochar in soil to justify your review of soilless properties and plant growth. 

Line 44: how much is a significant amount? Is this a personal observation or opinion? 

Lines 49-50: The references used to report the "damage to peat lands" where papers not specifically reporting research on peat lands but were general review or overview papers. In my opinion, to make sure strong accusations about the peat industry more current and peat specific works should be read and reported. There is a lot of debate of peat use around the world. 

Line 51: "limited supply such as bark". I question this and disagree. The paper cited is 12 years old and it not reflective of current bark supplies today. Caution should be made with current trends and issues and dated works on those topics. 

Line 64: hypothesis "that" (delete the)

I suggest maybe a section on Biochar Production Methods and a separate section of Biochar Feedstocks. More sections and subheadings may make the paper flow better and group topics together better. 

Lines 67-77: Why specifically mention and discuss these feedstocks and not all others in Table 1. In this one paragraph several different things are mentioned (randomly) feedstocks, some chemistry, nutrition, etc. Maybe start with the second paragraph on the three processes. 

Lines 75-76: Last statement seems out of place.

Lines 144-150: This discussion and the references seem to be on works conducted on biochar in soils. Are the authors inferring or extrappolating works in/on mineral soils to soilless container culture? There should be much caution to this. behaviour in soils cannot be assumed in organic substrates in pots.

Within Chemical Prop section, maybe add additional subheadings to help with organization and flow. pH, EC, CEC, etc.

Line 213: Adding biochar CAN cause the.....   (be careful to not make definitive statements).

Lines 222-250: Its interesting to separate out all the positive data reports and put them together and then in the next paragraph discuss the negatives. Not bad, just an interesting approach. Looking at (and reading) that paragraph the words that pop out noticeably are "positively, improved, enhanced, increased, etc." over and over. While I do not disagree that the reports being discussed made those positive claims about the effects/influence of biochar in those particular studied and circumstances, it does seem to give a biased approach to the reader regarding the "benefits" of biochar. After that paragraph the authors do detail the "less positive" aspects as well. 

The entire section on Effects of Biochar on Plant Growth is a very dense read. I know a review is a summary of many works completed on a subject but I am curious if more subheadings or separating the info in any way (further) could make it flow and read better instead of fact after fact being presented. 

Line 460: "certain conditions". I am pleased you mention this, because the response (positive or negative) of biochar in substrates depends heavily on management practice(s). Also on this line I would change "existing literatures" to "reviewed literature".

Lines 474-476: The statement about more research and use of biochar with compost is an opinion of the authors stated as a fact (in my opinion). I do not know how much "discussion" or non-referenced statements are allowed or acceptable in review articles but this seems to not add to the review as it was intended. Many in the industry (growers, researchers, industry manufacturers) believe the reason why composts and biochars are not major components of professional horticultural mixes....is because they are too variable and inconsistent. 

There is no doubt the tremendous effort and time spent in culminating this literature in the current form. It is a lot to digest and synthesize. A better outline or organization of it all would help the paper tremendously I think. Also paying close and careful attention to citing work (results) about topics based on papers that actually did the work....not papers that summarized the results over other works. For example, reference peat harvesting or sustainability science papers when discussing the impact of peat harvest on the environment or the shortage or available supplies of peat. Careful also to not extrapolate biochar data conducted in mineral soils trials with that of container substrates. Two different worlds, as you know.

Author Response

Reviewer #3:

Comments and Suggestions: Overall this review does not "flow well". I found it very choppy and difficult to read. Topics jumped around and the story was buried a bit. A lot of facts presented but the organization may could be improved.

We have improved our manuscript a lot by adding more quantitative results and critical evaluation of the results. Subheadings has been added to improve the organization based on the reviewer’s comments.

1.      Introduction: I would delete the first paragraph. This is a lot of references and space dedicated to biochar in soils, not in soilless substrates. I do not think it is necessary to overview biochar in soil to justify your review of soilless properties and plant growth.

The content about the use of biochar in soils in the introduction part has been deleted.

2.      Line 35: how much is a significant amount? Is this a personal observation or opinion?

That sentence has been deleted

3.      Lines 40-42: The references used to report the "damage to peat lands" where papers not specifically reporting research on peat lands but were general review or overview papers. In my opinion, to make sure strong accusations about the peat industry more current and peat specific works should be read and reported. There is a lot of debate of peat use around the world.

I have modified this part by choosing more current and peat specific reference.

4.      Line 43: "limited supply such as bark". I question this and disagree. The paper cited is 12 years old and it not reflective of current bark supplies today. Caution should be made with current trends and issues and dated works on those topics.

I have deleted this sentence.

5.      I suggest maybe a section on Biochar Production Methods and a separate section of Biochar Feedstocks. More sections and subheadings may make the paper flow better and group topics together better.

I have divide the section “Biochar Production” into two parts:

2.1. Biochar Production Methods

2.2. Biochar Feedstocks.

More subheadings have been added to this manuscript in all the other sections.

6.      Lines 142-143: Last statement seems out of place.

I have revised this paragraph and the last sentence to make it flow better.

7.      Lines 161-164: This discussion and the references seem to be on works conducted on biochar in soils. Are the authors inferring or extrapolating works in/on mineral soils to soilless container culture? There should be much caution to this. behavior in soils cannot be assumed in organic substrates in pots.

I have deleted the references on works conducted on biochar in soils in this section and have checked all the other contents. The references or works conducted in soil remained in this manuscript are only mentioned when talking about the original properties of the biochars themselves. All the references about the effect of biochar on container properties and plant growth are the works conducted in soilless container substrates.

8.      Within Chemical Prop section, maybe add additional subheadings to help with organization and flow. pH, EC, CEC, etc.

I have added subheadings.

9.      The entire section on Effects of Biochar on Plant Growth is a very dense read. I know a review is a summary of many works completed on a subject but I am curious if more subheadings or separating the info in any way (further) could make it flow and read better instead of fact after fact being presented.

I have added subheadings.

10.  The statement about more research and use of biochar with compost is an opinion of the authors stated as a fact (in my opinion). I do not know how much "discussion" or non-referenced statements are allowed or acceptable in review articles but this seems to not add to the review as it was intended. Many in the industry (growers, researchers, industry manufacturers) believe the reason why composts and biochars are not major components of professional horticultural mixes....is because they are too variable and inconsistent.

I have deleted the statement about the use of biochar with compost in this section.

All the other comments from Reviewer #3 have been addressed in the revised version and not mentioned here in detail.

Round 2

Reviewer 1 Report

I appreciate the effort of the authors improving the manuscript. However I still find that the manuscript is mostly descriptive and it is difficult to extract key/novel information that can be useful for the readers. I acknowledge the fact that the manuscript gathers lots of information (the most updated information) but it is not deeply evaluated. It is not clear the key messages that this manuscript will provide to the readers, probably in section 6 (presumably conclusions, rather than discussion).

Reviewer 3 Report

Major improvements were made to this manuscript. Authors took the comments from the review team seriously and worked to make positive corrections. The manuscript is better organized and easier to read. More focus solely on substrates and not mineral soil also improved this review.